# Bovine Coronavirus Prevalence and Risk Factors in Calves on Dairy Farms in Europe

**DOI:** 10.3390/ani14182744

**Published:** 2024-09-23

**Authors:** Anna Catharina Berge, Geert Vertenten

**Affiliations:** 1Veterinary Epidemiology Unit, Department of Internal Medicine, Reproduction and Population Medicine, Faculty of Veterinary Medicine, Ghent University, Salisburrylaan 133, 9820 Merelbeke, Belgium; 2MSD Animal Health, Wim de Körverstraat 35, 5831 AN Boxmeer, The Netherlands

**Keywords:** bovine coronavirus, preweaned calves, weaned calves, calf management, biosecurity

## Abstract

**Simple Summary:**

This study evaluated the prevalence and risk factors in health, management, and biosecurity of bovine coronavirus (BCoV) in neonatal and weaned dairy calves on 125 dairy farms in 17 countries in Europe. All farms had BCoV antibodies in the bulk tank milk. BCoV was detected by RT-PCR in 80% of dairy herds, and 24% of neonatal calves, 23% of weaned calves, and 4% of fresh cows sampled were shedding BCoV in nasal and/or fecal samples. The biosecurity on 109 dairies scored with Biocheck was, on average, 60% (external score 71%, internal score 47%), indicating that many farms are poorly protected from the introduction of BCoV onto a farm and spread within a farm. Dry cow vaccination against BCoV reduced shedding in neonatal calves, whereas it was linked to increased shedding in weaned calves in these farms. Several calf husbandry factors, such as transition milk feeding, milk feeding levels, group housing, and weaning age, were associated with BCoV shedding in calves.

**Abstract:**

This study evaluated prevalence and risk factors in health, management, and biosecurity of bovine coronavirus (BCoV) in neonatal and weaned dairy calves on 125 dairy farms in Europe. Nasal and fecal swabs from neonatal calves, weaned calves, and fresh cows were analyzed for BCoV using RT-PCR, and blood and bulk milk samples were collected for BCoV antibody levels using ELISA. Multiple logistic regression models with random effects of herds were used to evaluate the herd health status, husbandry, management, and biosecurity associated with BCoV shedding (nasal and/or fecal PCR positive samples) in neonatal and weaned calves. BCoV was detected in 80% of herds and in 24% of neonatal calves, 23% of weaned calves, and 5% of fresh cows. The biosecurity scored on 109 dairies with Biocheck.Ugent was, on average, 60% (external score 71%, internal score 47%), and there was no clear association between various biosecurity measures on BCoV shedding in calves. Dry cow vaccination against BCoV reduced shedding in neonatal calves, whereas it was linked to increased shedding in weaned calves in these farms. Several husbandry factors, including nutrition (transition milk feeding and milk feeding levels) and management (group housing and weaning age), were associated with BCoV shedding in calves.

## 1. Introduction

Bovine coronavirus (BCoV) is an enveloped positive sense RNA virus that is widespread around the world and causes bovine enteric or respiratory infections, leading to important economic losses. BCoV can be associated with calf diarrhea, bovine respiratory disease (BRD), winter dysentery, and combinations thereof [1,2]. Infection is primarily via fecal–oral and, to a lesser extent, respiratory (aerosol) routes, and the same strains can cause both respiratory and enteric symptoms [1]. Enteric BCoV infections are generally most severe in calves, with yellow to blood-stained mucus-containing diarrhea, which progresses to a profuse watery diarrhea [3]. Winter dysentery is a sporadic acute, contagious hemorrhagic enterocolitis of cattle that occurs in an epidemic fashion in a herd in cattle of various ages. BCoV is part of the BRD complex, and its role and impact remain to be further investigated [4,5,6]. 

Within a herd, reservoirs for BCoV infection may be virus cycling in clinically or subclinical infected animals. The transmission of enteric BCoV occurs from carrier dam to offspring postpartum or from infected calves housed in proximity to unexposed ones. Respiratory BCoV may be transmitted via the fecal–oral route and respiratory (aerosol) routes [7]. The evaluation of the geographic spreading pattern indicates two clusters: a European cluster and an American-Asian cluster. Epidemiological studies indicate a spread between herds associated with cattle movements but also through other vectors. Sequencing of isolates from Danish and Swedish dairies has indicated a varied reinfection within herds and transmission between herds [8].

There are a few studies of BCoV antibodies in individual milk samples and bulk tank milk samples. Bulk milk samples from 2236 Swedish dairy herds indicated 89% of herds were antibody-positive and 52% had high levels of antibodies to BCoV [9]. Swedish studies indicate that milk samples and bulk tank milk samples could be used for monitoring programs for antibodies to BCoV and indicated that milk samples from fresh primiparous cows could be screened [10,11]. Another Swedish study of primiparous dairy cows found BCoV milk antibodies in 77% of organic herds and in 85% of the sampled conventional herds [12]. In Norway, BCoV antibodies were found in bulk tank milk in 72% of 1347 dairy herds [13]. Bulk tank milk samples from 341 dairy herds in England and Wales indicated that 100% of herds had BCoV antibodies [14]. There are limited and not comparable studies of enteric and respiratory BCoV virus shedding in Europe [15,16,17,18,19,20,21,22]. 

There are studies indicating the importance of BCoV in both enteric and respiratory diseases in cattle and evidence that the virus is widespread in Europe. However, there is insufficient data to determine the spread of the virus in European dairy herds; the importance of biosecurity in BCoV prevention; and insufficient studies that evaluate herd management, health, and productivity factors for BCoV infection in calves. The aim of this study was to evaluate the prevalence of BCoV in neonatal and weaned calves on a representative set of European dairy farms and to determine the major biosecurity and management risk factors associated with enteric and nasal shedding of the virus in these calves. 

## 2. Materials and Methods

Farm enrollment: The aim of this cross-sectional field study was to have more than 130 dairy herds enrolled in European countries. The herd enrollment per country was approximately proportional to the national milk production. The herds were selected based on a convenience sample of 5–10 herds per country, identified by MSD Animal Health national technical teams and/or collaborating veterinary practitioners. The target minimum lactating herd size was 100 cows to have sufficient calves for sampling on one occasion. A derogation was made in Austria, where herds are, in general, small, and the herds were visited twice to obtain the targeted number of samples from calves. 

Sampling: On each dairy farm, 5 to 20 calves under one month of age (neonatal calves: NC), 5 to 20 calves within one month of weaning (weaned calves: WC), and 5–10 fresh cows within one month of calving (fresh cows: FC) were sampled. From each sampled animal, a rectal fecal and a nasal swab were collected with a cotton-tipped swab, which were immersed in tubes with a viral transport media (Sigma virocult MW950SENT, MWE, Wilts, UK), and a venous blood sample was collected from the jugular vein in calves and from the jugular or tail vein from cows. A bulk tank milk sample was collected in a sterile 50 mL container for BCoV antibody detection. 

Sample analysis: The samples were accompanied with a sampling record stating animal id; animal category (NC, WC, or FC); birth date; weaning date for WC and calving date for FC; and sampling date. Also, the animal health history within the previous 3 months was collected. The samples were sent chilled to the contracted laboratory in Boxmeer, the Netherlands. The presence of BCoV virus was analyzed with a duplex Real-Time RT-PCR (Kylt^®^ Bovine Coronavirus In-Vitro Diagnostica, AniCon Labor GmbH, Hoeltinghausen, Germany). Ct-values < 35 were classified as positive samples, and Ct-values ≥ 35 were classified as negative. Bulk tank milk BCoV antibodies and serum BCoV antibodies were analyzed using a competitive monoscreen ELISA test (Bio-X 392/2, Bio-X Diagnostics S.A., Rochefort, Belgium). The outcomes of the serum and bulk tank milk samples were percentage inhibition (%inh) of the optical density readings. The samples’ degree of positivity was categorized according to test manuals from 0 to 4 (%inh < 20 = 0, 20 ≤ %inh < 40 = 1 (+), 40 ≤ %inh < 60 = 2 (++), 60 ≤ %inh < 80 = 3 (+++), and %inh ≥ 80 = 4 (++++)).

Farm information collection: An extensive questionnaire was performed to determine the husbandry and management factors (Appendix A). Management and herd health history was collected with a focus on young stock diarrhea and respiratory disease prevalence, vaccines, treatments, and preweaning calf nutrition with focus on colostrum and milk or milk replacer usage. The investigator also asked over 120 questions on biosecurity according to the Biocheck.Ugent dairy scoring system (https://biocheckgent.com/, accessed on 19 September 2024) (Appendix A). All questionnaires and sampling records were transmitted to the principal study investigators through e-mail and hand-written records were transmitted with the samples to the contracted lab where all documents were scanned and transmitted electronically to the study investigator (G.V), who transmitted these further to the investigator (A.C.B) who evaluated documentation and submitted requests regarding missing or unclear data. 

Data analysis: All data were entered in Excel, and all statistical analysis was performed using SAS version 9.4. Data were categorized and/or made into binary answers, according to their distributions to facilitate statistical analysis. The sampled animals where BCoV was detected in either in feces and/or nasal swabs were categorized as ‘shedding BCoV’ or a ‘BCoV PCR-positive’ animal. Prevalence estimates for positive nasal and fecal samples and animals within categories, farms, and countries were calculated. Average antibody levels within animal categories, farms, and countries were calculated and correlated with the bulk tank antibody levels with Pearson correlation analysis. 

Biosecurity data analysis: The biosecurity questions were entered into the Biocheck.Ugent software for dairy and scored and categorized according to Biocheck.Ugent from 0 to 100% [23]. External biosecurity was composed of questions in 5 categories, including purchase and reproduction (BC-A), transports and carcass removal (BC-B), feed and water (BC-C), farm workers and visitors (BC-D), and control of vermin and pets (BC-E). The internal biosecurity was composed of 6 categories, including management of health (BC-F), calving (BC-G), calf (BC-H), milking parlor (BC-I), adult cattle (BC-J), and working and overall organization and equipment (BC-K). All categories have a subcategory score from 0 to 100% based upon weighted scores of questions, and these subcategories were further weighted and merged for scores for external, internal, and overall biosecurity according to the Biocheck algorithms. 

Statistical analysis: Univariable and multivariable analysis were used to determine the risks for BCoV in neonatal calves and weaned calves in a separate analysis. Calf category stratified univariable and multinomial logistic models were performed for neonatal and weaned calves, as they may have different risk factors and due to interaction effects between calf category and other predictive variables. All univariable analyses included a random effect of the farm (variables tested and direction of association and level of significance can be found in Appendix A). Four logistic regression models with random effects of herds (population-averaged marginal models) were used to determine the odds of neonatal and weaned calves that were BCoV-positive on PCR conditional on predictive variables (Proc Glimmix). In all these models, variables were selected for inclusion in the models using a stepwise procedure, where the *p*-value for entry was 0.3 and a *p*-value for retention was 0.1. Interaction effects were tested for all variables retained in the final models. Statistical significance was declared at *p*-value ≤ 0.05 and a non-significant trend at 0.05 < *p*-value ≤ 0.10. Confounding was evaluated by inspecting if the % difference is 10% or greater in the model with or without a confounding covariate. Two models evaluated biosecurity, management, and housing in relation to BCoV PCR-positive neonatal and weaned calves, including calf age as a covariate. In these models, Biocheck scores for the 11 individual biosecurity categories, as well as relevant individual questions, were statistically tested. Similarly, two multilevel logistic models evaluated the odds of BCoV PCR-positive neonatal and weaned calves on herd health factors, productivity, and antibody levels. 

## 3. Results

A total of 126 farms were included from April 2021 to December 2022. This study included dairy farms in Austria, Belgium, Czech Republic, Denmark, France, Greece, Hungary, Ireland, Italy, the Netherlands, Poland, Portugal, Romania, Sweden, Slovakia, and Scotland (United Kingdom) (Appendix A). The average lactating herd size was 370 cows, and 68% of herds were Holstein Friesian herds, and the remaining herds were either mixed breeds or other dairy breeds. There were, on average, 69 preweaned calves and 217 weaned non-pregnant calves. Sixty percent of the dairies reported using dry cow vaccinations against BCoV. 

A map describing the sampled farms (Figure 1, created in Google maps) shows clustered sampling in some countries. For seven farms, the biosecurity information was not completed. From an additional seven farms, the individual calf sample information was missing essential data, such as age at the time of sampling, thus excluding their use in the study. The data presented here were for neonatal calves from 113 farms and for weaned calves from 116 farms. The results from multilevel models may have a varying number of calves included due to missing explanatory variables.

### 3.1. BCoV Fecal and Nasal PCR-Positive

BCoV was detected in 27% of neonatal calves, in 24% of nasal samples, and 17% of fecal swabs (Table 1A). BCoV was detected in 32% of weaned calves, 21% in nasal and 22% in fecal swabs (Table 1B). Four percent of fresh cows were shedding BCoV in four percent of nasal and five percent of fecal swabs. BCoV was detected in 48% of 113 herds in neonatal calves (Figure 2) and in 71% of 116 herds in weaned calves (Figure 3). There was a relatively even distribution of percentage of neonatal and weaned calves per farm shedding BCoV (Figure 2 and Figure 3).

### 3.2. Bulk Tank Milk and Cattle BCoV Antibodies

Serum BCoV antibodies were detected in 92% of neonatal calves and 81% of weaned calves (at levels above 0 %inh in the ELISA). BCoV antibodies were detected in all bulk tank milk samples, the bulk tank milk (BTM) mean was 89 %inh (++++), and BTM antibodies in all farms were above 54 %inh (++) (Figure 4). There was a low correlation between bulk tank milk antibody levels and the animal antibody levels in the herds (R = 0.16). The shedding of BCoV in calves occurred in calves with all levels of antibodies (Figure 5).

### 3.3. Biosecurity Associated with BCoV in Calves (NC and WC)

The biosecurity was scored with Biocheck. The scoring tool was used for 118 farms, and the average Biocheck score was 60% (external score 71%, internal score 48%). The distribution of the Biocheck scores for the categories A–K are described in Figure 6. 

### 3.4. Management and Biosecurity Associated with BCoV in NC and WC

The logistic regression evaluating herd management and biosecurity factors associated with BCoV in neonatal calves included 1332 calves, of which 355 calves were shedding BCoV in 110 dairies (Table 2). This model indicates that the odds of BCoV shedding in neonatal calves increased 2.3 times for every subsequent week the first month of life. The odds of BCoV shedding in calves decreased by 55% in dairy herds that used dry cow BCoV vaccination. The odds of BCoV shedding in neonatal calves was 2.5 times higher with group housing starting the first week and 1.9 times higher the second week compared to the third week or later. There was a 41% reduction in odds of BCoV shedding in calves that received transition milk feeding the first week. The odds of BCoV shedding were 2.6 times higher in herds where the milk feeding volume the first month was less than 6 L/day compared to farms that fed more than 6 L/day. In this model, an increasing Biocheck B score was associated with higher odds of BCoV shedding in calves, whereas, with every increasing percentage of replacement heifers purchased in the previous year, there was a 25% increase in the odds of neonatal BCoV shedding. 

The logistic regression evaluating herd management and biosecurity factors associated with BCoV in weaned calves included 1459 calves, of which 472 calves were shedding BCoV in 116 dairies (Table 3). This model indicates that the odds of BCoV shedding in weaned calves decreased 15% for every increasing month of life. The odds of BCoV shedding in calves was 1.81 times higher in dairy herds that used dry cow BCoV vaccination and was 2.24 times higher in herds using bovine respiratory disease (BRD) vaccines in calves compared to non-vaccinating herds. The odds of BCoV shedding in weaned calves was reduced by 30% in herds where weaning was, on average, prior to 10 weeks of age and reduced by 45% in herds where group housing was practiced prior to 8 weeks of age. None of the biosecurity measures implemented were significantly associated with BCoV shedding in weaned calves. 

### 3.5. Herd Health Factors Associated with BCoV in NC and WC

The logistic regression evaluating calf and herd health, and the productivity factors associated with BCoV in neonatal calves included 1346 calves, of which 342 calves were shedding BCoV in 112 dairies (Table 4). This model indicates that the odds of BCoV shedding in neonatal calves increased 90% for every increasing week of life (weeks 1 to 4) and with increasing the BCoV antibody levels (0–4), the odds of shedding decreased 24%. The odds of BCoV shedding were 54% lower in herds with cow average yearly (305 day) milk production above the median of the herds and 68% higher in herds with BTM somatic cell counts above the median of the herds. The odds of BCoV shedding increased 1% for every % increase in the reported herd average % of calves that had diarrhea in the preweaning period. The odds of BCoV shedding were highest in herds reporting that diarrhea started predominantly in the first week of life and 66% lower in herds reporting diarrhea starting in week two. The odds of BCoV shedding in calves was 2.5 times higher in herds reporting a BRD outbreak in the previous year. In herds where BCoV shedding was detected in fresh cows, the calves were 4.1 times more likely to shed BCoV.

The logistic regression evaluating calf and herd health and productivity factors associated with BCoV in weaned calves included 1410 calves, of which 459 calves were shedding BCoV in 113 dairies (Table 5). This model indicates that the odds of BCoV shedding in weaned calves decreased 15% for every increasing month of life (months 2–5), and with increasing the BCoV antibody levels, the odds of shedding increased 16%. The odds of BCoV shedding were 39% higher in herds with milk production above the median of the herds. The odds of BCoV shedding increased 1% for every % increase in the reported herd average % of calves that have diarrhea in the preweaning period. The odds of BCoV shedding in calves was 34% higher in herds reporting a BRD outbreak in the previous year. In herds where BCoV shedding was detected in the fresh cows, the calves were 2.5 times more likely to shed BCoV. There was also a significant but small increase in the odds (1.001) of BCoV shedding in herds, with an increasing number of weaned non-pregnant heifers. 

## 4. Discussion

This study has shown that BCoV is commonly present in both the respiratory and enteric pathway in neonatal and weaned calves in European dairies, with all study herds being seropositive to the virus, and the virus was present in 80% of the study herds. Within the herds, 27% of preweaning calves, 32% of weaned calves, and 4% of fresh cows were shedding BCoV (BCoV-positive fecal and/or nasal samples).

The study could not evaluate country-level prevalences due to clustered sampling in some countries, which is a weakness due to the convenience enrollment of farms. We also intended to sample large countries such as Germany and England, however, the national regulatory authorities had enrollment criteria and costs that were not compatible with this study. There were no visible changing trends in BCoV prevalence in calves from north to south or west to east in Europe. This comes not as a surprise, as numerous studies have shown the global widespread nature of BCoV in dairy cattle [24]. Studies in Asia (China and Thailand) indicates similar endemic situations as described in our study [25,26]. 

There are previously limited and not comparable studies of enteric and respiratory BCoV in Europe. A study of French dairy calves in 1983–1984 found coronavirus in 42% of the samples and strangely found it more prevalent in healthy calves compared to diarrhea calves [15]. A study of 65 dairy herds in Spain found BCoV in 20% of samples from calves shedding rotavirus [16]. BCoV was detected in 26% of 230 fecal samples from preweaned calves in 100 Austrian farms [17]. In Switzerland, BCoV was detected in 8% of samples from diarrheic dairy calves on 71 dairies [18]. In Sweden, BCoV was found in 3% samples from calves, and limited BCoV virus was found in less than 200 normal or diarrheic samples in Norway [19,20]. Published data on respiratory BCoV are sparse. A Belgian cross-sectional study of 128 BRD outbreaks (dairy, mixed dairy, and beef farms) in 2016 to 2018 indicated that BCoV was the most frequently isolated virus (38%) [5]. A study in Ireland of nasal swabs from 1364 BRD outbreaks found BCoV in 23% of the samples [21]. A Polish study of young stock indicated a seroprevalence of 73% and a PCR virus detection in 11% of the animals sampled [22]. A study of water buffalo and dairy cattle in the Campania region in Southern Italy similarly found high seroprevalence (31% of the animals sampled) [27]. 

Scandinavian studies have indicated that bulk tank milk serology can provide an estimate of the herd-level seroprevalence of BCoV [10,28]. However, our study has indicated that bulk tank milk BCoV antibody levels are of limited value to evaluate animal antibody levels or the level of BCoV infection in animals, and this could be due to the endemic situation in Europe and possibly also that serum levels in fresh cows and neonatal calves are impacted by dry cow vaccinations. 

We found similar levels of nasal and fecal shedding in weaned calves, whereas nasal shedding was higher in preweaned calves. A clinical trial by Cho et al. indicated that, regardless of route of exposure, oral or nasal, BCoV was detected in nasal and fecal samples from all infected calves and that nasal shedding occurred prior to fecal shedding regardless of the inoculation route, and the fecal shedding persisted longer [29]. We noted this in our study too, where, in the second week, nasal shedding was twice as high as fecal shedding. 

As it is important to know herd-level risk factors for BCoV to effectively prevent and control the virus, this study aimed to identify herd-level risk factors. Biosecurity is very important to reduce the spread of diseases between and within farms. A Swedish study found a clear association between higher herd biosecurity levels and a lower prevalence of BCoV herd infection [30]. However, in Sweden, they evaluated antibody levels, while, in our study, we focused on BCoV detected with PCR in nasal and fecal swabs. In this cross-sectional study, it is important to note that we can only evaluate associations; however, causal factors cannot be determined. It should be noted that most dairy farms do not have high levels of biosecurity and that partial measures may still lead to the dissemination of BCoV between and within farms. One of the major risks for the introduction of contagious pathogens is through live animal introduction [31]. In the neonatal calves, BCoV shedding increased with an increasing percentage of replacement heifers purchased. Higher biosecurity for farm workers and visitors (BC-B category) was found associated with increased shedding in neonatal calves. This may be confounding, since higher biosecurity for people may be practiced due to higher infectious disease pressure on the calves in farms. It has been shown that BCoV can persist on clothes, boots, and equipment up to 24 h after exposure to virus-shedding calves [32], and not providing boots for visitors was a risk factor for BCoV seropositivity in dairy herds in Sweden [33]. In Norway, there is a stakeholder voluntary preventive program against BRSV and BCoV in cattle herds, with antibody classification of herds and enhanced external biosecurity [34]. In the Norwegian BCoV control program, it was noted that BCoV introduction through professional visitors, such as AI personnel, could be prevented with good biosecurity measures. Unfortunately, the prevention of the introduction of BCoV through people is not practiced sufficiently in European dairies, and many of the study farms indicated that veterinarians and other professional visitors, such as inseminators and claw trimmers, that all come in direct contact with the animals, did not use farm-dedicated protective clothing and boots. With the high prevalence and an endemic situation of BCoV in dairy farms in Europe, it is likely that partial biosecurity is not effective in endemic situations with highly infectious viruses. Furthermore, there are studies indicating that neonatal calves infected with BCoV can become persistently infected [35].

For husbandry/management factors associated with BCoV shedding in neonatal calves, we noted that using transition milk feeding and higher milk or milk replacer feeding volumes could reduce BCoV shedding. There are now numerous studies showing the importance of transition milk or colostrum supplementation the first week of life on gut health [36]. The importance of a good nutritional status for optimal immunity [37] is important to emphasize in disease prevention programs. It was noted that group housing starting in the first and second weeks compared to after the third increased the shedding of BCoV in neonatal calves. This is likely due to increased contacts between the calves in the group housing systems. In weaned calves, there was a lower likelihood of BCoV shedding in systems where group housing started prior to 8 weeks of age and calves were weaned prior to 10 weeks of age. The initiation of group housing after or in combination with weaning is stressful for previously single-housed calves, and this grouping stress combined with increased contact with herd mates may have led to increasing the numbers or intensity of BCoV shedding. This study cannot determine why the delayed weaning was associated with a higher likelihood of BCoV shedding, as it was also noted that, with increasing months of age in the weaned calves, there was a reduced risk of BCoV shedding. In a study in Thailand, the only factor that was found to be protective against being seropositive was disinfecting diarrheal stools compared to doing nothing or simply washing away stools [25]. This practice is very rare in Europe. In a Dutch study, non-optimal hygienic housing was also identified as a risk factor for BCoV shedding in neonatal calves [38]. 

Our study showed a reduced level of shedding in neonatal calves with increasing antibody levels, and this is mostly colostral-derived BCoV antibody protection of the calves under 1 month of age. However, in weaned calves, shedding was higher in calves with higher antibody levels, and this is most likely indicating a mounting immunity due to active BCoV infections in the calves. In this study, 60% of farmers used a dry cow vaccine against calf enteric pathogens, including BCoV, to boost colostral immunity in the calves. This vaccination was shown to be effective in reducing BCoV shedding in neonatal calves. This was contrary to a clinical trial in Korea, where they concluded that “The effectiveness of maternal antibodies in preventing viral replication and shedding appeared limited” [6]. However, there was increased shedding in weaned calves in our study herds using BCoV dam vaccination. This could have been due to lower infection rates in the neonatal calves, reducing the calves’ natural acquired immunity to infection post-weaning. At the time of the study, the nasal vaccine against BCoV that is currently on the market in the EU was not available for farmers. The involvement of BCoV in the BRD complex was and is still to a large degree underestimated by veterinary practitioners and calf owners. In our study, BCoV shedding was much higher in weaned calves in herds using BRD vaccination. There is increasing evidence of BCoV’s role in BRD, and studies have shown that BCoV is also in the lower respiratory tract of cattle with respiratory disease [39]. It may thus be likely that BCoV was contributing to the BRD challenges, to which the farm was vaccinating against other respiratory pathogens. The use of vaccines for BCoV will likely continue to evolve, as it has been shown to be a valuable tool against various coronaviruses in multiple species [40].

Herd health and productivity factors were evaluated separate from management and nutrition factors, as these are associated, and including these factors together in one model may create confounding. In both neonatal and weaned calves, BCoV shedding was more common in herds where fresh cows were detected to be shedding BCoV. It could not be determined whether the positive results in the fresh cows where the result of reactivation of a persistently infected animal [35] or a recent infection in the fresh cows. BCoV shedding was also more common in neonatal and weaned calves in herds that reported a BRD outbreak in the previous year. There were some unexplainable associations with average cow yearly milk production in that it was associated with reduced shedding in neonatal calves and increased shedding in weaned calves. BCoV shedding in neonatal calves was also more common in dairy herds with BTM somatic cell counts above the median.

In our study, we noted that neonatal and weaned calves were more likely to shed BCoV in herds where diarrhea was more commonly seen in the preweaning period. This was also noted in a Swedish study where BCoV endemic and recently infected herds had more coughing and diarrhea in young stock and cows compared to BCoV free herds [41]. A study in the Netherlands evaluated 424 calves under 22 days of age from 108 dairy farms and found that *E. coli* K99 was more likely to be detected in feces from calves where BCoV has been detected [38]. A study in veal calves in the Netherlands similarly noted that calves shedding BCoV were more likely to have more severe diarrhea days, and this was coupled with a reduced weight gain in the preweaning period [42]. Since higher preweaning weight gain has been associated with higher first lactation milk yield, the negative production economic consequences of BCoV on dairies span beyond the treatment costs and immediate losses [43]. This study confirms that there are clear associations between BCoV being actively shed by calves on dairies and the herd-level calf health. 

Our study adds to the body of evidence of the widespread distribution of BCoV in dairy calves. BCoV is associated with increased calf health challenges, such as severe neonatal diarrhea and calf respiratory disease. The economic losses due to the morbidity and mortality of calves, reduced growth performance, and decreased milk production of adult dairy cattle [2,41,42] indicate that BCoV cannot be disregarded in sustainable health dairy production. 

## 5. Conclusions

This study has shown that BCoV is commonly present in both the respiratory and enteric pathway in neonatal and weaned calves in European dairies, with all study herds being seropositive to the virus, and the virus was present in 80% of the study herds. The overall biosecurity levels on dairies in Europe, in general, show weaknesses. It is likely that, in endemic situations, such as this one, the most important measures to reduce the disease impacts of BCoV on calves include optimizing immunity through management and vaccinations. BCoV dam vaccinations have a good protective effect on neonatal calves but cannot protect the weaned calves. Several health and management factors might reduce the risk of BCoV infections such as transition milk feeding, high daily milk volumes to calves, and grouping of calves after 3 weeks of age but prior to weaning. 

## Figures and Tables

**Figure 1 animals-14-02744-f001:**
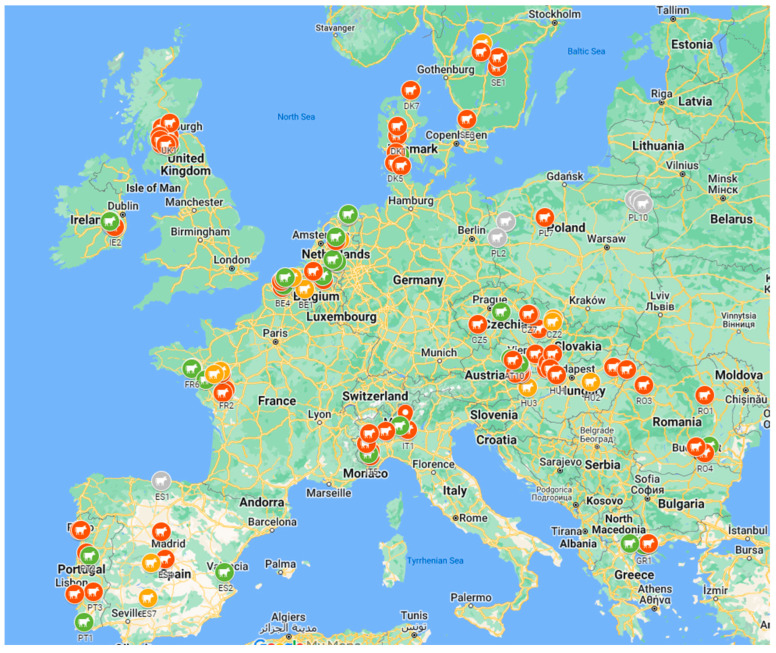
Map of dairy farms included in this BCoV prevalence and risk factor study. The coloring of farms indicates levels of BCoV shedding in calves; green—0%, 0% < orange < 10%, red ≥ 10%, and gray—excluded farms.

**Figure 2 animals-14-02744-f002:**
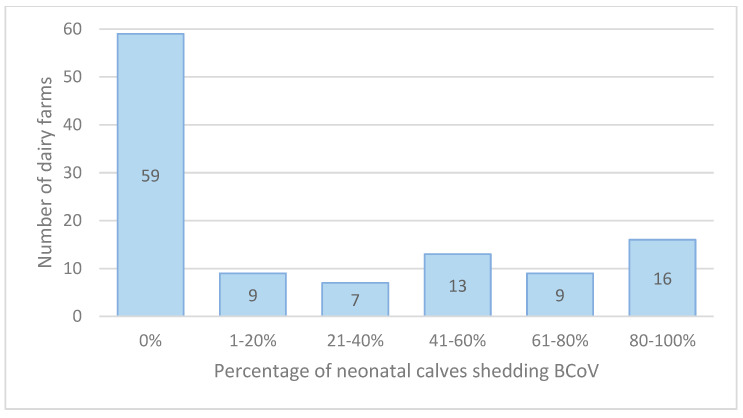
Number of herds (of 113) stratified by percentage of neonatal calves shedding BCoV.

**Figure 3 animals-14-02744-f003:**
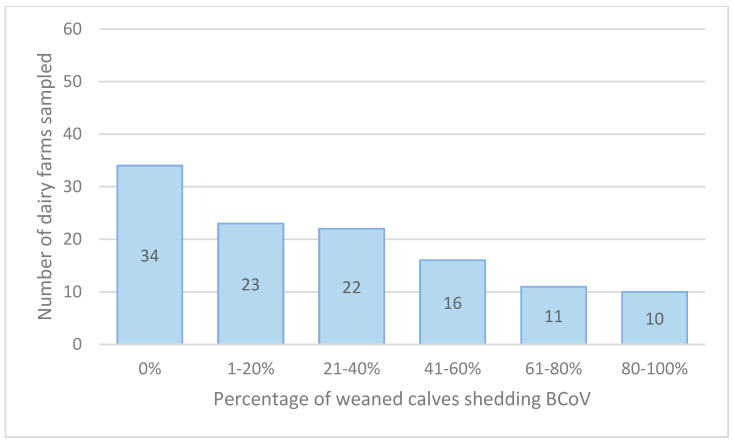
Number of herds (of 116) stratified by percentage of weaned calves shedding BCoV.

**Figure 4 animals-14-02744-f004:**
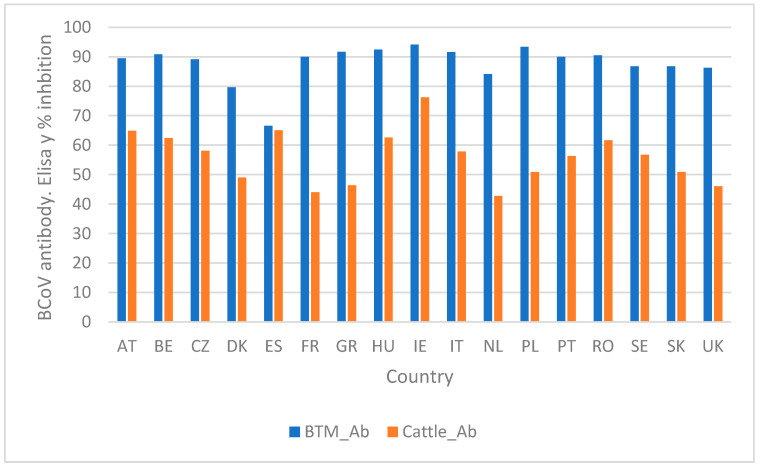
BCoV mean antibody levels (% inhibition in Elisa) in bulk tank milk (BTM_Ab) and the mean antibody levels of neonatal calves, weaned calves, and fresh cows (Cattle_Ab) in the enrolled European dairy farms stratified by country.

**Figure 5 animals-14-02744-f005:**
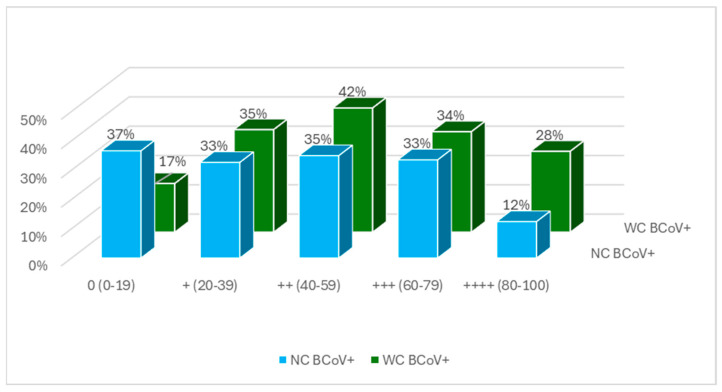
Percentage of calves shedding BCoV stratified by BCoV antibody levels in neonatal (NC BCoV+) and weaned (WC BCoV+) calves.

**Figure 6 animals-14-02744-f006:**
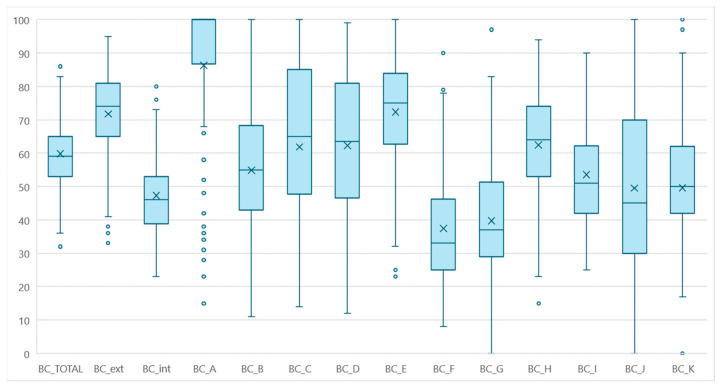
Boxplot distributions of Biocheck scores for 118 dairies. BC-ext (External biosecurity) includes BC-A (purchase and reproduction), BC-B (transports and carcass removal), BC-C (feed and water), BC-D (farm workers and visitors), and BC-E (control of vermin and pets). The BC-int (internal biosecurity) includes BC-F (management of health), BC-G (calving management), BC-H (calf management), BC-I (milking parlor), BC-J (adult cattle management), and BC-K (working and overall organization and equipment).

**Table 1 animals-14-02744-t001:** BCoV detection in nasal and fecal swabs from neonatal calves (A) and weaned calves (B) on European dairies.

	Neonatal Calves BCoV Recovery *				Weaned Calves BCoV Recovery *		
A	Fecal+	Fecal−	Sum (%)	B	Fecal+	Fecal−	Sum (%)
Nasal+	196	124	320 (24%)	Nasal+	167	145	312 (21%)
Nasal−	40	994	1034 (76%)	Nasal−	160	984	1144 (79%)
Sum (%)	236 (17%)	1118 (83%)	1354	Sum (%)	327 (22%)	1129 (78%)	1456

* −/+ indicate detection of BCoV with RT-PCR.

**Table 2 animals-14-02744-t002:** Logistic regression model describing management and biosecurity factors associated with odds of shedding BCoV in neonatal calves.

					95% Conf. Int.		
Variable	Param.	Est.	S.E	OR	Lower CI	Higher CI	*p*-Value
Intercept	cont.	−3.79	0.36	0.02	0.01	0.05	<0.01
Week of life (1–4)	cont.	0.83	0.08	2.29	1.96	2.68	<0.01
Dam BcoV−	Yes	−0.80	0.16	0.45	0.33	0.61	<0.01
vaccination	No	0.00	0.00	1.00	1.00	1.00	ref
Biocheck score B (%)	cont.	0.02	0.00	1.02	1.02	1.03	<0.01
% Heifers purchased	cont.	0.23	0.08	1.25	1.07	1.47	<0.01
Group housing	week 1	0.92	0.19	2.51	1.73	3.65	<0.01
started	week 2	0.64	0.21	1.90	1.26	2.86	<0.01
	week 3+	0.00	0.00	1.00	1.00	1.00	ref
Transition milk	Yes	−0.52	0.21	0.59	0.39	0.90	<0.01
feeding	No	0.00	0.00	1.00	1.00	1.00	ref
Milk/MR	<6 L	0.97	0.18	2.63	1.85	3.74	<0.01
daily quantity	≥6 L	0.00	0.00	1.00	1.00	1.00	ref

**Table 3 animals-14-02744-t003:** Logistic regression model describing management and biosecurity factors associated with the odds of shedding BCoV in weaned calves.

					95% Conf. Interval		
Variable	Param.	Est.	S.E	OR	Lower CI	Higher CI	*p*-Value
Intercept	cont.	−0.69	0.28	0.50	0.29	0.86	0.01
Month of life	cont.	−0.16	0.07	0.85	0.74	0.98	0.03
Dam BCoV	Yes	0.59	0.14	1.81	1.37	2.38	<0.01
vaccination	No	0.00	0.00	1.00	1.00	1.00	ref
BRD calf	Yes	0.80	0.13	2.24	1.73	2.89	<0.01
vaccination	No	0.00	0.00	1.00	1.00	1.00	ref
Calf weaning age	<70 days	−0.35	0.13	0.70	0.55	0.91	<0.01
	≥70 days	0.00	0.00	1.00	1.00	1.00	ref
Group housing	<56 days	−0.60	0.13	0.55	0.43	0.71	<0.01
started	≥56 days	0.00	0.00	1.00	1.00	1.00	ref

**Table 4 animals-14-02744-t004:** Logistic regression model describing calf and herd-level health, and productivity factors associated with the odds of shedding BCoV in neonatal calves.

					95% Conf. Interval		
Variable	Param.	Est.	S.E	OR	Lower CI	Higher CI	*p*-Value
Intercept	cont.	−2.30	0.29	0.10	0.06	0.18	<0.01
Week of life (1–4)	cont.	0.62	0.08	1.87	1.61	2.17	<0.01
Antibody level (0–5)	cont.	−0.27	0.05	0.76	0.69	0.85	<0.01
Ave 305d	≥median	−0.78	0.15	0.46	0.34	0.62	<0.01
cow milk yield	<median	0.00	0.00	1.00	1.00	1.00	ref
BTM SCC	≥median	0.52	0.15	1.68	1.25	2.27	<0.01
	<median	0.00	0.00	1.00	1.00	1.00	ref
% Calf diarrhea preweaning	cont.	0.01	0.00	1.01	1.01	1.02	<0.01
Primary week	week 1	0.00	0.00	1.00	1.00	1.00	ref
of diarrhea	week 2	−0.83	0.16	0.44	0.32	0.60	<0.01
	week 3	−0.26	0.22	0.77	0.50	1.18	0.17
BRD outbreak	yes	0.90	0.18	2.47	1.73	3.51	<0.01
previous year	no	0.00	0.00	1.00	1.00	1.00	ref
Cow BCoV	yes	1.42	0.15	4.13	3.05	5.58	<0.01
shedding	no	0.00	0.00	1.00	1.00	1.00	ref

**Table 5 animals-14-02744-t005:** Logistic regression model describing calf and herd-level health, and productivity factors associated with the odds of shedding BCoV in weaned calves.

					95% Conf. Interval		
Variable	Param.	Est.	S.E	OR	Lower CI	Higher CI	*p*-Value
Intercept	cont.	−1.30	0.26	0.27	0.16	0.45	<0.01
Months of life	cont.	−0.16	0.08	0.85	0.73	0.98	0.03
Antibody level (0–5)	cont.	0.15	0.05	1.16	1.05	1.27	<0.01
Ave 305d	≥median	0.33	0.13	1.39	1.09	1.79	0.01
cow milk yield	<median	0.00	0.00	1.00	1.00	1.00	ref
% Calf diarrhea preweaning	cont.	0.01	0.00	1.01	1.00	1.01	<0.01
BRD outbreak	yes	0.30	0.16	1.34	0.98	1.85	0.07
in previous year	no	0.00	0.00	1.00	1.00	1.00	ref
Cow BCoV shedding	yes	0.93	0.13	2.54	1.98	3.28	<0.01
in previous year	no	0.00	0.00	1.00	1.00	1.00	ref
Number of weaned non-pregnant heifers	cont.	0.00	0.00	1.001	1.00	1.00	0.02

## Data Availability

The data are not publicly available due to privacy restrictions.

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
