# Peer review of "Bovine Coronavirus Prevalence and Risk Factors in Calves on Dairy Farms in Europe"

_animals, 2024, doi:10.3390/ani14182744_

Round 1
Reviewer 1 Report
Comments and Suggestions for Authors
The present study is a large-scale investigation regarding the prevalence of BCoV in Europe (several countries have been included in the research). The topic is of great interest, and the sections are well developed and described. I suggest minor revisions as listed above:
1) Line 13: The authors should also specify the total number of sampled farms. Moreover, they should specify the individual prevalence (both serological and molecular).
2) Line 23-26: Too much similan to "simple summary.".
3) Line 83: Other studies were performed in Italy (Campania, seroprevalence of 30.8%), Thailand (98%), and Ghana (55%). I suggest including these studies. Some of them also performed a risk factor analysis that could be useful for the discussion section.
4) Line 93: In my opinion, this section should be divided into subsections, for example: sampling, molecular analysis, serological analysis, statistical analysis.
5) Line 432: As stated previously, the authors should discuss their results with those described in literature (all the studies that performed risk analysis).
Comments on the Quality of English LanguageThe paper is clear and well-written. Only small errors have been detected.
Author Response
The present study is a large-scale investigation regarding the prevalence of BCoV in Europe (several countries have been included in the research). The topic is of great interest, and the sections are well developed and described. I suggest minor revisions as listed above:
Dear reviewer, thank you for the time and expertise that you have contributed to our review. Please find reply to your comments in cursive below.
1) Line 13: The authors should also specify the total number of sampled farms. Moreover, they should specify the individual prevalence (both serological and molecular).
Unfortunately, the max word count for the simple summary of 200 words prevents this detail. We clearly state this in the body of the paper.
2) Line 23-26: Too much similan to "simple summary.".
We have revised simple summary and abstract.
3) Line 83: Other studies were performed in Italy (Campania, seroprevalence of 30.8%), Thailand (98%), and Ghana (55%). I suggest including these studies. Some of them also performed a risk factor analysis that could be useful for the discussion section.
The Thailand study by (Ven et al., 2021).has been referenced and identified risk factors from this study have been discussed ’.The Italian study in Campania has been referenced and reviewed in discussion (Ferrara et al., 2023) . There is a study in Ghana, where cattle of non-specified age and production category have been sampled, and a very low virus recovery under 1% was seen. This reference is not included, since the sampling strategy in relation to prevalence cannot be evaluated (Burimuah et al., 2020).
4) Line 93: In my opinion, this section should be divided into subsections, for example: sampling, molecular analysis, serological analysis, statistical analysis.
Sub-sections have been entered for the Material and Methods section.
5) Line 432: As stated previously, the authors should discuss their results with those described in literature (all the studies that performed risk analysis).
We have discussed the results of this study with the other relevant publications that have performed risk factor analysis. Please see answer to point 3.
Reviewer 2 Report
Comments and Suggestions for Authors
Please see attached file.

Author Response
Minor correction 179 The were → ??? 278 66% → 56% ?
I have reviewed the sentence. The odds ration 0.44 indicates indicates a 56% lower odds, and the odds ratio 1.66 indicates a 66% higher odds. Thus, I believe the sentence is correctly reflecting the statistical model.
The description of References list is not unified especially in page number such as 643220, 349-64, 147-152 and so on. 147-152 is best. Please check again and correct.
In this first submission I left the reference software formatted references, facilitating revisions. The references have been formatted and updated with page number correctly.
Reviewer 3 Report
Comments and Suggestions for Authors
This manuscript investigates the prevalence and risk factors of bovine coronavirus (BCoV) in calves on European dairy farms.
The content from lines 65 to 82 belongs in the discussion section and should be moved there. The introduction should focus solely on the topic and the study at hand.
The Materials and Methods section needs to be detailed in a scientific manner, not superficially. Please divide the section into subsections such as: study design, sample collection, viral RNA screening or RT-PCR procedure, antibody screening, statistical analysis, etc. This will allow readers to easily refer back to specific methods when interpreting the results.
Please include a table under the Results section listing the farms, their locations, the number of samples collected, and all relevant data related to the samples.
The results mention that BCoV was detected in 27% of neonatal calves and 32% of weaned calves, but these numbers cannot be found in Table 1A or 1B.
Figure 2 is difficult to understand, and the legend is insufficiently detailed.
Figure 3 is similarly unclear. The results mentioned in the text are not reflected in the figure.
Regarding the statement: "BCoV antibodies were detected in 92% of neonatal calves and 81% of weaned calves at levels above 0," please clarify from which samples these data were obtained and what assay was performed.
For Figure 4, what do the abbreviations under the bar graphs represent? Are they referring to countries?
The statement, "There was a low correlation between bulk tank milk antibody levels and the animal antibody levels in the herds (R = 0.16)," should specify the data or methods that this correlation is based on.
The meaning of "The shedding of BCoV in calves occurred in calves with all levels of antibodies (Figure 5)" is unclear and requires rephrasing for clarity.
There are several other areas in the draft that lack clarity and detail.
Author Response
Dear reviewer. Thank you for your valuable review. Please find below responses to your comments in Italics.
The content from lines 65 to 82 belongs in the discussion section and should be moved there. The introduction should focus solely on the topic and the study at hand.
This section has been moved to the discussion.
The Materials and Methods section needs to be detailed in a scientific manner, not superficially. Please divide the section into subsections such as: study design, sample collection, viral RNA screening or RT-PCR procedure, antibody screening, statistical analysis, etc. This will allow readers to easily refer back to specific methods when interpreting the results.
The material and methods section have been modified to include sub-sections with sub-section titles.
Please include a table under the Results section listing the farms, their locations, the number of samples collected, and all relevant data related to the samples.
An additional Table S3 has been created with details regarding the sampled farms per country, the samples taken from neonatal calves, weaned calves and fresh cows and the & of animals BCoV shedding per animal category adn the mean antibody levels per animal category.
The results mention that BCoV was detected in 27% of neonatal calves and 32% of weaned calves, but these numbers cannot be found in Table 1A or 1B.
These results can be verified in Table 1A and 1B by adding up calves found positive in at least one sample (196+124+40)/1354 for neonatal calves and (167+145+160)/1456 for weaned calves.
Figure 2 is difficult to understand, and the legend is insufficiently detailed.
The title has been changed to ’ Figure 2. Number of herds (of 113) stratified on percentage of neonatal calves shedding BCoV
Figure 3 is similarly unclear. The results mentioned in the text are not reflected in the figure.
The title has been changed to ’ Figure 3. Number of herds (of 113) stratified on percentage of weaned calves shedding BCoV
A sentence has been added in text to further describe pictures ’ There was a relatively even distribution of percentage of neontal and weaned calves per farm shedding BCoV (Figure 2 and 3).
Regarding the statement: "BCoV antibodies were detected in 92% of neonatal calves and 81% of weaned calves at levels above 0," please clarify from which samples these data were obtained and what assay was performed.
The sentence has been modified to ’ Serum BCoV antibodies were detected in 92% of neonatal calves and 81% of weaned calves (at levels above 0 %inh in the ELISA).
For Figure 4, what do the abbreviations under the bar graphs represent? Are they referring to countries?
In figure 4, the legend of the x-axis states ’Country’. The abbreviations are the standard country codes. We are willing to add the country codes in text, according to Editors recommendations.
The statement, "There was a low correlation between bulk tank milk antibody levels and the animal antibody levels in the herds (R = 0.16)," should specify the data or methods that this correlation is based on.
We have added in material and methods that we used Pearson correlation analysis.
The meaning of "The shedding of BCoV in calves occurred in calves with all levels of antibodies (Figure 5)" is unclear and requires rephrasing for clarity.
We elaborate on the BCoV shedding in neonatal and weaned calves in relation to their antibody levels based on the logistic models in Table 4 and 5. We do not want to go into trends on shedding based on Figure 5, but prefer to this based upon the multivariable analysis.
There are several other areas in the draft that lack clarity and detail.
We have reviewed the whole paper again, and hopefully some of the areas that are not clear to you and the other reviewers and future readers have been addressed.
Round 2
Reviewer 3 Report
Comments and Suggestions for Authors
The draft has been improved. Thank you for taking the time to address my comments.